# Dynamically Updated Alive Publication Date

**Mikhail Gorbunov-Posadov** 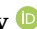

Information and Publishing Department, Keldysh Institute of Applied Mathematics, 125047 Moscow, Russia; gorbunov@keldysh.ru

**Abstract:** A scientific work posted on the internet, which its authors constantly keep up to date, is called an 'alive' publication. The genre of alive publishing has many attractive features. However, it requires a certain expansion of the composition of the meta-attributes of the publication: along with the traditional attributes, the date of the appearance of the new, fresh revision is brought to the fore here. Such a date is placed in a prominent place in the text of the publication. Along with this, it becomes highly desirable to include a dynamically ("on the fly") generated date in a bibliographic reference to an alive publication. The currently used methods of dynamic extraction of this date are considered for a simple online publication, for a publication that has received a DOI through Crossref, and for a publication posted in arXiv. Thanks to adding this meta-attribute, references to alive publications will improve any bibliographic list.

**Keywords:** alive publication; dynamic component of bibliographic reference; latest revision date; Crossref; arXiv





## 1. Introduction

Until recently, the proverb "Littera scripta manet" ("The written word endures" in Latin) dominated everywhere, including in the world of science. A mistake made in a published article was almost impossible to correct; it haunted the author for the rest of their life and confused readers. In addition, obtaining any new results in the field under study each time required the author to issue a new article, and where necessary to devote a very significant part of the text to repeating previously published information, without which a fresh reader could not receive this new information.

The advent of the internet and the subsequent transition of the mass reader of scientific publications online make it possible to leave these annoying circumstances in the past. Now any online article in a couple of minutes can be replaced with its corrected and/or expanded version. The author who does not take advantage of this happy opportunity not only deprives readers of the chance to find out the latest news from the area that interests, but also often forces readers to deal with a text containing detected errors.

## 2. Materials and Methods

A scientific work posted on the internet, which its author constantly keeps up to date, is called an alive publication [1]. Due to their obvious advantages, alive publications are gaining more and more supporters every year.

- An author who has abandoned traditional, static publication in favor of the alive form finds themself in a new, significantly more comfortable and productive environment. The mistakes and typos made are no longer fatal; they can be easily corrected. The circle of readers of an alive publication is much wider. Interest in them often even increases over time: many readers return to their favorite text over and over again, not only to refresh their memory of the most significant moments but also to find out how the author's views have evolved and what new trends are suggested or noted from others researchers in field under consideration.

- For the reader, an alive publication is undoubtedly preferable to a static one. Indeed, how much more confident do you feel when you know that the text in front of your eyes is under the vigilant control of the author; that all inaccuracies and errors noticed since the first posting of the work online have been carefully corrected in it; that the text constantly reflects the changes taking place in the branch of science under consideration.

Of course, it would be utopian to imagine the life of the author of an alive publication cloudless. In particular, serious difficulties arise here in connection with the usual official requirements of reporting on publications. Scientific reporting is often calculated exclusively in items, and in this case, in the eyes of an official, an alive publication seems to be a reckless challenge, an emphatically irrational expenditure of creative efforts.

The use of the term "alive" publication is not conventional practice. The terms "living" [2,3], "evolving" [4] "dynamic" [5], "liquid" [6,7], "propelled", "movable", "progressing", "developing", and "advancing" are used more often. To our regret, most of these terms generally mean that the publication contains multimedia and/or interactivity rather than alive content.

### 2.1. Reviewing

First of all, we note that online conservation of an error that can be corrected in a couple of minutes is a crime against science. This is a fire that is better put out urgently and only then should the author deal with the related formalities.

Is it necessary to obediently wait for the reviewer's approval in the case when the changes are not of the nature of extinguishing a fire, but only improve the existing text? Of course, if the reviewer is always at hand, then why not? Unfortunately, this is not usually the case. The reviewer is often far away and has no time to deal with your edits, and to distract the reviewer with a request to authorize the insertion of a missing comma, of course, is ridiculous. However, the cursed comma often haunts the author both day and night.

However, there is a way out, and not just one.

First, the corrected version of the article can be presented in preprint status that peacefully coexists with the peer-reviewed version. In this case, the decision is up to the reader: you can read the version sanctified by the reviewer, but probably outdated and containing uncorrected errors, or can read the latest version—a preprint. It is easy to predict which of the versions most readers will choose here.

Another solution is offered by the Ridero Publishing House [8], which produces books according to the "print on demand" scheme. Ridero allows the author to change the content of the published peer-reviewed book, but this is allowed no more than once a quarter. Thus, the load of reviewers is reasonably regulated. Otherwise the load could be prohibitive due to agitated authors who change their text a thousand times.

Another solution is possible, by analogy with the F1000 Publishing House [9]. The article is reviewed and posted on the publisher's website. Next, the author changes the text, which is constantly publicly available. In particular, any of several thousand staff reviewers of the publishing house can spontaneously become acquainted with the updated text and write a review. After the author takes into account the comments contained in the review, the new version of the article receives the status "reviewed".

Finally, if the changes were so extensive that it is already possible to talk about a new article, then the author can publish this new version in the same or in a new journal. Anyway, it is relatively easy to combine an alive publication and its peer-reviewed version.

### 2.2. Logging

How significant is the logging of all changes made to the alive publication? Mandatory full logging leads either to an uncontrolled expansion of the volume of protocols, or to artificial containment of the author who, for reasons of protocol economy, is forced to postpone current changes until a lot of them accumulate. Both options are flawed, each in its own way.

Who might be interested in the history of the changes being made? A strictly compliant and documented complete history of changes is necessary, for example, for an online archive serving a current legislature. In addition to observing the necessary legal rigor, here the visitor to the archive should also be given the opportunity to familiarize themself with the legislative norms that were in force at a particular time of interest. However, it is obviously not necessary to document exhaustively any comma added or excluded by the author from an ordinary scientific article.

Developers of large-scale software systems such as Microsoft Windows or Visual Studio are forced to assemble a group of changes to a large package. Here, the installation of changes requires the user's attention, and constantly tormenting the user with small installations would, of course, be inhumane. Only sometimes a small but urgent installation is required; due, for example, to the need to promptly fend off a rapid epidemic of a dangerous virus. The size of the package of corrections and additions is often very impressive; it can be accompanied by a digest briefly describing the changes being made.

In the case of a scientific article, the protocols are rather the sphere of interest of only a historian of science, but not an ordinary reader of the article for the first time. However, the reader who returned to the article after some time, of course, could be briefly informed about what happened to the text during this absence. The opportunity provided to the reader to subscribe to changes is useful here, when the introduction of noticeable changes is always accompanied by the mailing of a digest letter.

At the same time, a specific technical solution for logging changes in an alive publication is not such an urgent problem. For example, in arXiv [10] all changes made are logged, but there is no subscription option. Nonetheless, everything has been working successfully for many years: both authors and readers have become used to it, and adapted.

## 3. Results

### 3.1. Date of Last Update

How can a reader distinguish an alive publication from a static one? Simply adding a special "Publication declared alive" icon to its representation is obviously not enough. After all, the author could once insert this icon and forget about it and about the online text. Therefore, the only reliable evidence of an alive publication is the fresh date of its last update. This date is certainly shown in a prominent place in the main text, for example, in the form of a conspicuous banner (Figure 1).

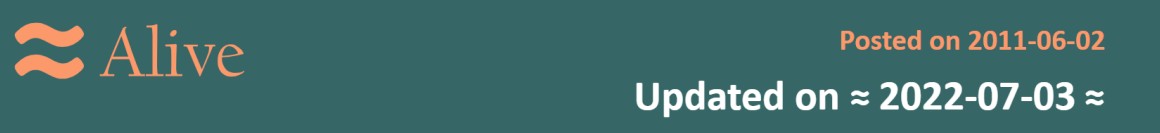

**Figure 1.** Banner of an alive publication. The date of the first placement and the date of the current (fresh) revision are indicated.

The banner containing the date serves as a reliable guide for the reader of the alive publication. Such information is undoubtedly useful, but it is equally important to inform the reader that the publication is alive in another common situation—when viewing a reference to it in the bibliographic list.

In bibliographic references to online materials, such dates as "last modified" or "accessed" are often found. These dates are static and therefore unsuitable for servicing an alive publication. Their appearance is because the author foresees a change in the material under consideration. Such a date only tells the reader that at a certain moment the quoted material was contained at the specified address, and the author is not responsible for what happened to it afterwards.

*3.2. Dynamically Updated Date*

The reader looking through bibliographic references is not indifferent about which of the listed publications are alive and which are "dead" or static, i.e., they have not changed since their first appearance. At the same time, marking a publication in the bibliographic list as "alive" is not enough here: as mentioned above, the author could have forgotten many years ago about their intention to turn to this genre. Then, such a mark would lose its meaning; it would simply disorient the reader. Hence, the only reliable evidence of the "living" of the publication is the presentation to the online reader of the fresh date of its last edition.

To implement such a presentation, the HTML format is desirable. If relatively recently the aging PDF format was the main and practically the only means of online presentation of scientific publications, now HTML, thanks to its numerous advantages, is gradually gaining a stable position in this area. One of the advantages of HTML is the ability to include dynamically generated elements in the publication text relatively easily.

Therefore, in order to attract the attention of the reader of the bibliographic list to alive publications, software tools were developed for HTML [11,12]. These tools implement cross-domain relations and allow them to dynamically supplement the usual text of the bibliographic reference with a new important component—the fresh date of the last revision of the publication.

To do this, a special construction is added to the source HTML text of the bibliographic reference, serving the formation of the final text of the reference presented to the online reader with the updated date. By means of this construction, when forming the text of the reference, the file of the alive publication is accessed which, generally speaking, is hosted in another, external domain. The update date is contained there in a certain format in the attributes of the alive publication, extracted from there "on the fly" and included in the text of the bibliographic reference presented to the online reader.

Thus, the online reader of the bibliographic list always sees the date of appearance of the new version of the alive publication, which is really the latest at the moment. For presentation in the bibliographic reference of the dynamic date of updating, we use the prefix "Updated on", and surround the date itself with the characters "≈", for example:

M. Gorbunov-Posadov. Alive publication // Open systems. 2011, № 4. P. 48–49. (In Russian). Updated on ≈ 2022-07-03 ≈ https://keldysh.ru/gorbunov/live.htm

The hyperlink included in the bibliographic reference here leads precisely to the latest revision of the publication.

*3.3. Crossref*

The hyperlink included in the bibliographic reference, due to well-known technological considerations, now increasingly leads not directly to the publication file, but turns to the DOI (digital object identifier). In most cases, the DOI is supplied by the Crossref agency. For a long time, Crossref has held a tough position towards alive publications: it was written in the agency's rules that no changes could be made to a publication that had received a DOI.

However, over time, alive publications in Crossref were fully legalized. On the one hand, the requirement of the immutability of the received DOI material was excluded from the rules. On the other hand, the Crossmark mechanism was implemented, assuming that all newly appearing versions of an alive publication coexisted with their predecessors, and each of them receiving its own DOI. In each of these versions, the "Check for updates" icon serving the alive publication (Figure 2) is placed in a prominent place.

**Figure 2.** Crossmark icon serving alive publication in Crossref.

Crossmark does not allow us to include directly in the bibliographic reference a permanent hyperlink to the latest version of an alive publication. Instead, it is proposed to include a hyperlink to the current version at hand in the bibliographic reference. However, a reader who has somehow wandered into such a possibly outdated version, by clicking on Crossmark icon, can find out if and where a more recent version of an alive publication exists and is located, as well as whether this publication has been retracted by the editors.

The Crossref's proposed mechanism for serving alive publication by Crossmark seems inefficient. Access to the latest version through an outdated one is unnatural. In addition, the reader may simply not pay attention to the Crossmark icon and thus not guess the existence of a more recent version of the publication. Finally, even if it were somehow possible to find out the date of the revision of the latest version, it is undesirable to include this date in the bibliographic reference to the outdated version because the reader may mistake it for the updated date of the old version. However, without specifying a fresh date, a link to an alive publication is boring, and in a certain sense even incorrect.

At the same time, Crossref does not insist that Crossmark is the only possible approach to versions maintenance. It allows not only its own DOI for each version (as in Crossmark), but also one DOI for all versions, which in this case replace each other under this address. Both of the approaches, according to Crossref [12], have advantages and disadvantages.

In our opinion, preference should be given to the approach with only one DOI. It is more productive and more comfortable for the reader that an external link, in this case, a DOI hyperlink, always leads directly to the latest, most fresh version of the material. In other words, let the subsequent versions of the alive publication replace each other under the same DOI. However, for a lover of antiquity on the page of the alive publication, you can provide a link to the protocol of alive publication changes that is being formed and stored somewhere aside.

With such a service organization, the corresponding bibliographic reference can and should be supplemented with a dynamically updated date of the last revision of the alive publication [13]. It is reliable evidence that the author does not forget about the constant support of this production. For example:

M. Gorbunov-Posadov. Online bibliographic reference //
Keldysh institute preprints. 2020. № 11. Updated on $\approx$ 2022-07-19 $\approx$
https://doi.org/10.20948/prepr-2020-11

If the author plans to re-index each new version of an alive publication (of course, under the same DOI) in Crossref, then the date declaration in the alive publication file can be omitted. If the date in this file is not declared, the date of the last indexing of the publication in Crossref will be inserted as the update date.

At the same time, if the changes made only affected the main text—i.e., neither the location of the file, nor the bibliography, nor the abstract, nor other meta-attributes have changed—then, generally speaking, you can save a little effort—post the next version, but not index it in Crossref again. In this case, to serve the dynamic date included in the bibliographic reference, the publication file will need to explicitly specify the date of the last revision, as in the case of a direct (without DOI) link to an alive publication.

*3.4. arXiv*

arXiv [10] is the oldest and largest archive of scientific preprints placed in the public domain. It has been operating since 1991, and by 2022 more than two million preprints were placed in arXiv.

arXiv supports hosting alive publications. The author has the right to place new, and new versions, of an alive preprint in arXiv at any time, the files of which receive addresses (URLs) with suffixes v1, v2, v3, ... (Figure 3).

**Figure 3.** Four versions of the alive publication presented on the page https://arxiv.org/abs/1710.0 2185 (accessed on 1 September 2022).

To access the latest (fresh) version of an alive publication in arXiv, a URL without a suffix is used, for example, https://arxiv.org/abs/2103.10761 (accessed on 1 September 2022). A bibliographic reference using such a shortened URL can be supplemented with a dynamically updated posting date in the arXiv of the latest version [14], for example:

Gorbunov-Posadov M.M. Alive publication.
Revision from ≈ 2021-03-19 ≈
https://arxiv.org/abs/2103.10761

In this case, it is not necessary to declare the date of the last revision in the publication file: the date is extracted from the arXiv system data.

## 4. Discussion

Alive publication, due to its obvious advantages, is steadily gaining acceptance in the scientific community. When the officials making organizational decisions finally recognize the significant scientific merit of the author not only publishing an article in an authoritative publication, but also the up-to-date support of an alive publication, then the distribution of alive publications will acquire a massive character, which will undoubtedly benefit modern science.

The inclusion in the bibliographic reference of the dynamic date of the last revision of an alive publication requires relatively little effort from the author. For the reader, such a date turns out to be extremely useful; transitions from the bibliographic list to recently updated alive publications are usually performed many times more often than to static ones that are not provided with a dynamic date.

**Funding:** This research received no external funding.

**Data Availability Statement:** Not applicable.

**Conflicts of Interest:** The author declares no conflict of interest.

**Disclaimer:** This paper has been reviewed following a rigorous peer review process and is published on its merits, and that this decision was reached by the editorial board following open discussion.

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
