# Peer review of "Dynamically Updated Alive Publication Date"

_publications, doi:10.3390/publications10040048_

Round 1

Reviewer 1 Report

The topic and concept are interesting, and I think there is potential here, but I wish the article had engaged a bit more with some of the challenges of this model. For instance, if an author chooses to revise/update a publication, does it need to go through peer review again before these changes are published? How does this factor into the workload of editors and reviewers? Are changelogs maintained or access to earlier versions for those wishing to see what revisions were made? (You did touch a bit on this last point but mostly within the context of DOIs.) Additional data would really help to make this case. The manuscript also needs to be revised to strengthen the clarity of the English text, which might also help to resolve some of the concerns cited above, because at points it was difficult to follow the argument as currently written. With some significant editing and a bit of additional research to flesh out the pros and cons of the "alive publication" model to accompany your argument that the pros outweigh the cons, I think this could be a compelling article.

Author Response

Dear colleague,

Thank you for your positive assessment and for the helpful feedback!

It seemed to me that the issues of reviewing and logging lie somewhat apart from the main topic of the article. However, both reviewers spoke in favor of more detailed coverage of these issues. Apparently, this really should be done. In the new version of the article, reviewing and logging of alive publication are discussed in the new sections "2.1. Reviewing" and "2.2. Logging".

I tried to clarify some of the wording. If the editorial board finds it useful I will, of course, turn to the editing services provided by the journal.

Reviewer 2 Report

Very interesting article!

I recommend changing all of the he/his to they/them to be more gender inclusive.

I recommend minor edits on language use. 

My major concern relates to lack of discussion in terms of tracking the changes that are made. Your article does not discuss if there is a way for readers to see what changes are made over time, if there is such a way. The article could be improved by adding if there is this capability for alive articles, and if so, how that works and how those changes can be viewed by readers. If it's not a feature, maybe a discussion on why it's not featured or a possibility to be added to enhance alive publications. 

Author Response

Dear colleague,

Thank you for your positive assessment and for the helpful feedback!

I excluded from everywhere the words he, his, him. Unfortunately, I am absolutely not guided in this area and do not understand why this should be done.

I tried to clarify some of the wording. If the editorial board finds it useful, I will turn to the editing services provided by the journal.

It seemed to me that the issues of logging lie somewhat apart from the main topic of the article. I only briefly touched on this issue in section 3.3. However, both reviewers favor more detailed coverage of these issues. Apparently, this really should be done. In the new version of the article, logging of alive publication is discussed in the new section "2.2. Logging".

Round 2

Reviewer 1 Report

The article is certainly improved from its initial draft, and I am glad to see the new sections included, although I wish the logging section struck for more of a happy medium -- it's certainly not necessary to track every minor edit such as a missed comma or a corrected misspelling, but any significant content changes should be noted in some form for the benefit of those citing a work. However, I do realize that this is more of a conceptual overview and so the specifics of how this would work are outside the scope of this article. This is more of a focus on why we should use alive publication, not how it should necessarily work.

The only significant concern I have is that the writing could still use some additional work to improve the clarity of the text. If the writing can be improved, then I have no other significant reservations about the article.

Author Response

Dear colleague,

You are right. The problem of maintaining correct citation is too multifaceted, not always related to alive publication and does not fit into my note. After all, in the limit for such a service, when quoting, it is always necessary to keep a complete copy of the cited work somewhere independently, because otherwise the citation looks unreliable: the work may not be online later.

I apologize for the fairly noted low quality of my text. I will try to work on it some more.

Thank you for the benevolent attitude to the article.

Reviewer 2 Report

It looks to me that all comments were incorporated. 

Author Response

Thank you so much for your benevolent attitude